# Molecular mechanisms of re-emerging chloramphenicol susceptibility in extended-spectrum beta-lactamase-producing Enterobacterales

Fabrice E. Graf [1] ✉, Richard N. Goodman [2], Sarah Gallichan[1], Sally Forrest[1], Esther Picton-Barlow [1], Alice J. Fraser[2], Minh-Duy Phan [3,4,5], Madalitso Mphasa[6], Alasdair T. M. Hubbard [2,7], Patrick Musicha[6], Mark A. Schembri [3,4,5], Adam P. Roberts [2], Thomas Edwards [2], Joseph M. Lewis [1,6,8] & Nicholas A. Feasey [1,6,9]

Infections with Enterobacterales (E) are increasingly difficult to treat due to antimicrobial resistance. After ceftriaxone replaced chloramphenicol (CHL) as empiric therapy for suspected sepsis in Malawi in 2004, extended-spectrum beta-lactamase (ESBL)-E rapidly emerged. Concurrently, resistance to CHL in *Escherichia coli* and *Klebsiella* spp. decreased, raising the possibility of CHL re-introduction. However, many phenotypically susceptible isolates still carry CHL acetyltransferase (*cat*) genes. To understand the molecular mechanisms and stability of this re-emerging CHL susceptibility we use a combination of genomics, phenotypic susceptibility assays, experimental evolution, and functional assays for CAT activity. Here, we show that of 840 Malawian *E. coli* and *Klebsiella* spp. isolates, 31% have discordant CHL susceptibility genotype–phenotype, and we select a subset of 42 isolates for in-depth analysis. Stable degradation of *cat* genes by insertion sequences leads to re-emergence of CHL susceptibility. Our study suggests that CHL could be reintroduced as a reserve agent for critically ill patients with ESBL-E infections in Malawi and similar settings and highlights the ongoing challenges in inferring antimicrobial resistance from sequence data.

Antimicrobial resistance (AMR) is a major threat to global health. Drug-resistant bacterial infections were estimated to be associated with 4.95 million deaths in 2019, with sub-Saharan Africa being the worst affected region[1]. Among the most problematic drug-resistant bacteria are extended-spectrum beta-lactamase-producing Enterobacterales (ESBL-E) which are resistant to 3rd-generation cephalosporins (3GCs) and classified as priority pathogens by the WHO[2]. ESBL-E infections are associated with higher morbidity, mortality and economic burden in

[1]Department of Clinical Sciences, Liverpool School of Tropical Medicine, Liverpool, UK. [2]Department of Tropical Disease Biology, Liverpool School of Tropical Medicine, Liverpool, UK. [3]Institute for Molecular Bioscience (IMB), The University of Queensland, Brisbane, QLD, Australia. [4]School of Chemistry and Molecular Biosciences, The University of Queensland, Brisbane, QLD, Australia. [5]Australian Infectious Diseases Research Centre, The University of Queensland, Brisbane, QLD, Australia. [6]Malawi-Liverpool Wellcome Research Programme, Kamuzu University of Health Sciences, Blantyre, Malawi. [7]Department of Biosciences, School of Science and Technology, Nottingham Trent University, Nottingham, UK. [8]Institute of Infection, Veterinary and Ecological Sciences, University of Liverpool, Liverpool, UK. [9]School of Medicine, University of St Andrews, St Andrews, UK. ✉e-mail: fabrice.graf@lstmed.ac.uk

the treatment of bloodstream infections in Malawi[3,4], and in low resource settings such as Malawi access to effective treatment alternatives (e.g. carbapenems, or beta-lactam/beta-lactamase inhibitor combinations) is often lacking. The prevalence of ESBL-E rapidly increased in Malawi after the 3GC ceftriaxone replaced chloramphenicol (CHL) as first-line empiric therapy for suspected sepsis from 2004[5,6].

CHL is an inhibitor of protein synthesis and broad-spectrum antibiotic discovered in 1947[7], which is now rarely used, primarily because of the risk of severe adverse effects[8]. These include dose-unrelated, aplastic anaemia, which is irreversible and fatal[9] but rare (incidence between 1:19,000 and 1:270,000)[10], dose-related, reversible bone marrow suppression[11] and Grey Baby Syndrome[12]. As effective and safer broad-spectrum beta-lactam antibiotics were introduced, they replaced CHL in most settings. The dominant mechanism of CHL resistance is mediated by CHL acetyltransferases (CAT) inactivating CHL by acetylation[13]. Other mechanisms include efflux pumps, inactivation by phosphotransferases, target site mutations and decreased membrane permeability[13].

While resistance to 3GCs, fluoroquinolones and aminoglycosides increased in Malawi, resistance to CHL decreased as its use declined[5]. *Escherichia coli* and *Klebsiella pneumoniae* populations showed decreasing proportions of CHL-resistant isolates, from around 80% in 1998–2004 to around 50% and below in 2012, sparking an interest in whether CHL can be re-introduced as a treatment option for ESBL-E infections in critically ill patients where there is no alternative therapy[5,14]. Some bacteria isolated in Malawi, however, have a discordant CHL genotype–phenotype, i.e. they still harbour CHL resistance genes despite being phenotypically susceptible[15–18]. Therefore, simple CHL resistance gene loss caused, for example, by lineage or plasmid replacement in the bacterial population is unlikely. With the prospect of using CHL as a potential reserve agent for 3GC-resistant infections it is important to understand (i) the molecular mechanism(s) of this re-emerging CHL susceptibility, i.e. the molecular basis of CHL susceptibility genotype–phenotype mismatches, (ii) the stability of the phenotypic susceptibility, and (iii) how widespread this phenomenon is.

In this work, we investigate a collection of ESBL *E. coli* and *K. pneumoniae* with CHL susceptibility genotype–phenotype mismatches. Using functional assays in combination with genomic data, we determine the CHL sensitivities, CAT enzyme activity, and functional expression of *cat* genes, to understand the molecular mechanism of re-emerging CHL susceptibility. Further, we test the stability of CHL susceptibility in several isolates employing experimental evolution with increasing concentrations of CHL, use co-occurrence analysis of AMR genes and investigate the spread of *cat* alleles in the context of sequence types to identify potential drivers to explain the high proportion of CHL susceptibility in ESBL-E populations in Malawi.

## Results

### Genotype–phenotype mismatch for CHL resistance

We screened a collection of 566 *E. coli* and 274 *K. pneumoniae* species complex (*KpSC*) isolates previously isolated from patients admitted to Queen Elizabeth Central Hospital (QECH) in Blantyre, Malawi. All isolates had been whole-genome sequenced, of which 164 (93 *E. coli*, 71 *KpSC*) were from sentinel surveillance of bacteraemia[5,15,16] and 676 isolates (473 *E. coli*, 203 *KpSC*) were asymptomatically carried ESBL-isolates from stool, collected in a research study[14,17,18] (Supplementary Data 1). Of the total isolates, 53.5% (449/840) were phenotypically susceptible to CHL (Fig. 1a) and 31.0% (260/840) had a discordant genotype–phenotype. 43.9% (197/449) of phenotypically susceptible isolates harboured CHL resistance genes; the majority were CHL acetyltransferase (*cat*) genes (188/197) (Fig. 1b). Overall, the most prevalent *cat* genes were *catB4* (229) followed by *catA1* (189), *catA2* (104) and *catB3* (19) (Fig. 1c). Other known CHL resistance genes, *cmlA1*

(29), *cmlA5* (23) and *floR* (14), were less common in the collection and their presence correlated well with phenotypic resistance (Supplementary Fig. 1). We selected a subset of 42 isolates, 27 *E. coli* (13 CHL-R, 14 CHL-S) and 15 *K. pneumoniae* (6 CHL-R, 9 CHL-S), based on a genotype–phenotype mismatch, i.e. CHL phenotypically susceptible isolates harbouring diverse *cat* genes and we included isolates without a genotype–phenotype mismatch with a matched *cat* gene as controls (same sequence type (ST) where available) for further in-depth functional analysis (Table 1) to investigate the molecular mechanism of those mismatches. First, we determined the CHL susceptibility for each isolate by broth microdilution and compared the result to previously collected AST data[5,14], which was determined for CHL using the disc diffusion method. For 81.0% (34/42) isolates, broth microdilution confirmed the result of disc testing and we concluded that phenotype-genotype mismatches were not explained by inaccurate phenotypic data, however, initial numbers of genotype–phenotype mismatches (Fig. 1) might have been over- or underestimated. Five of the eight isolates that had discordant disc diffusion and MIC results were within one doubling dilution of the breakpoint for CHL (>8 μg/mL). We used the EUCAST (v.12.0) MIC breakpoint >8 μg/mL to classify isolates as resistant or susceptible. Four isolates were reclassified as susceptible and four as resistant.

### No CAT activity in susceptible isolates

Next, we tested all 42 isolates for CAT enzyme activity to determine whether *cat* genes were functionally expressed. We adapted the rapid CAT assay (rCAT)[19] as an indirect measure of enzyme activity. The rCAT assay measures free sulfhydryl groups of the CAT substrate and acetyl donor acetyl-coenzyme A in cell lysates and a colour change to yellow indicating CAT enzyme activity can be spectrophotometrically measured.

All phenotypically CHL-S isolates, irrespective of the presence of a *cat* gene, were negative for CAT activity (Fig. 2). All but five resistant isolates (CAC10A, CAC122, CAC10H, CAB199, and CAC13Y) with a *cat* gene showed CAT activity. Three of those CHL-R/rCAT negative isolates (CAB119, CAC10H, and CAC13Y) carried *catB4* genes and this was the only *cat* gene present. Isolate CAB199 (CHL MIC = 128 μg/mL) co-carried a *floR* gene which could contribute to the CHL-R phenotype. CAC10A had a *catA1* in addition to *catB4* but a weak CHL-R phenotype (MIC = 16 μg/mL). We used the disc diffusion (dCAT) assay[20] as a secondary and independent assay, which is based on the ability of CAT-producing bacteria to cross-protect susceptible bacteria by inactivating CHL. Tested isolates (CAB199, CAC13Y) failed to cross-protect a sensitive strain, thus confirming that *catB4* in those isolates is non-functional (Supplementary Fig. 2). One isolate (CAC122) that carried *cat_pC221*, a *cat* gene originating from *Staphylococcus aureus*[21], did not show enzyme activity and its resistance phenotype is likely explained by *cmlA1*. This *cat* gene was shown to be inducible and regulated through translational attenuation[22]; we thus tested this isolate with a sub-MIC concentration of CHL in culture media but no CAT activity was observed. We concluded that resistant isolates with *catA1*, *catA2* and *catB3* have functional CATs whereas isolates with *catB4* do not.

Of 23 phenotypically susceptible isolates, 15 carried *cat* genes (three with *catA1*, two with *catA1* + *catB4*, and ten with *catB4*), none of these showed CAT activity, suggesting that *cat* genes in those CHL-S isolates are not expressed.

### Functional expression of *catB4* does not confer CHL resistance

To assess the effect of the genetic context of *cat* genes in our isolates, we cloned the coding sequence (CDS) of the dominant *cat* variants present in the collection (*catA1*, *catA2*, *catB3*, and *catB4*) into the expression vector pEB1-*sfGFP*[23] under the constitutive *proC* promoter, by replacing *sfGFP*, and tested their susceptibility to CHL in a clean genetic background of *E. coli* MG1655. MG1655 pEB1-*catB4* had an MIC of 8 μg/mL identical to the MG1655 pEB1-sGFP control whereas all other

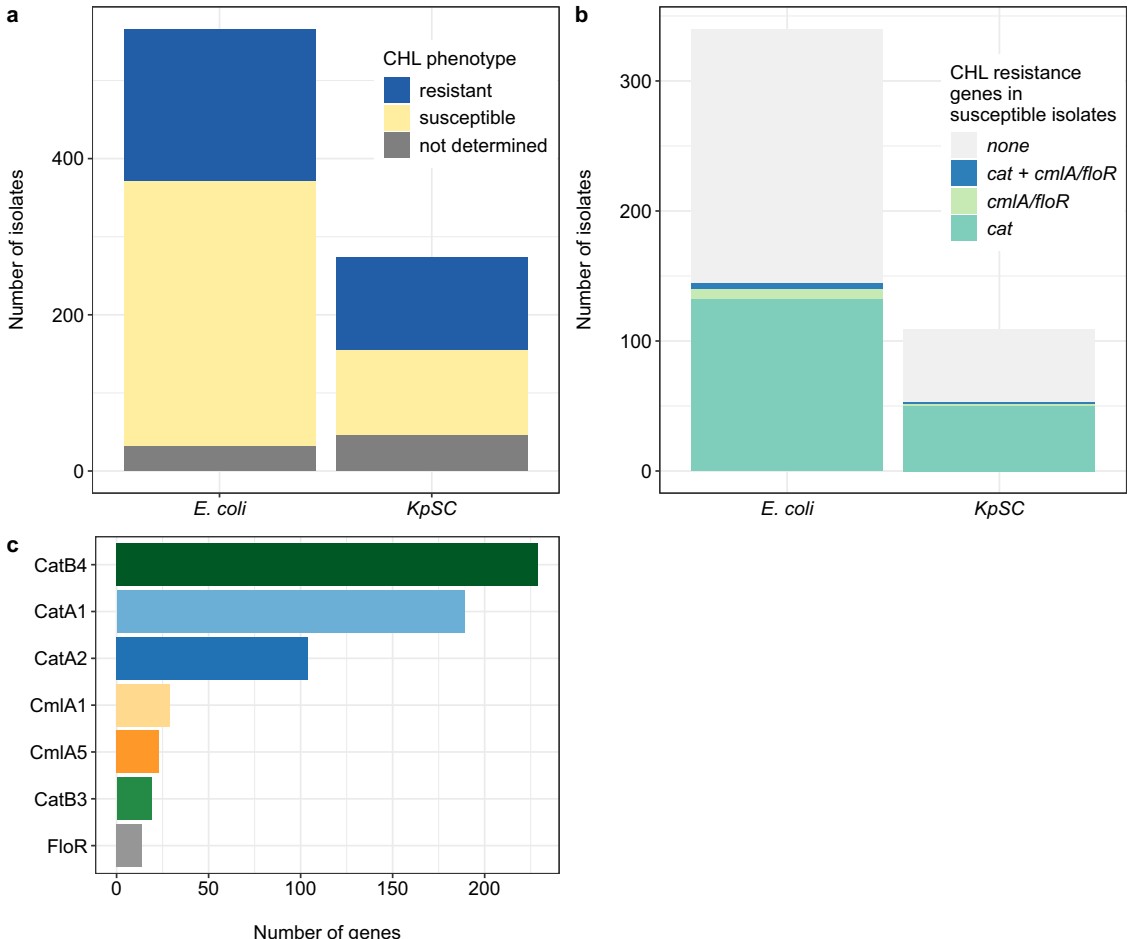

**Fig. 1 | Characteristics of isolates. a** Phenotypic CHL susceptibility of *E. coli* and *KpSC* isolates based on disc diffusion. **b** Number of CHL susceptible isolates carrying *cat*, *floR* or *cmlA* genes, both (*cat* + *floR* or *cmlA*) or no CHL resistance genes. **c** Number of CHL resistance genes present in the 840 Malawian isolates. Source data are provided as a Source Data file.

*cat* variants had an MIC of >512 μg/mL confirming that *catB4* is a non-functional variant.

### *catB4* is a non-functional truncated *catB3*

Functional CAT assays (rCAT and dCAT) demonstrated that *catB4* in the genetic context of tested isolates was non-functional and ectopic expression of *catB4* confirmed that the CDS does not produce a functional product that confers CHL resistance. We previously observed that in addition to a full-length assembled *catB4* (549 bp), many isolates had several partial assemblies in their genomes of approximately 107 bp (Supplementary Fig. 3a and ref. 24). We investigated the 107 bp (position 443–549 of *catB4*) by BLASTn against NCBI's nucleotide collection and obtained many hits among different Enterobacterales with 100% query cover and sequence identity matching the "IS6-like element IS26 family transposase". We next compared *catB4* with *catB3* by pairwise sequence alignment. Both CDS share 100% sequence identity for the first 442 bases and only differ after position 443, corresponding to the last 107 bp of *catB4* matching with the IS26 sequences (Fig. 3a). All but seven *catB4* assemblies in our isolate collection share 100% sequence identity (Supplementary Fig. 3b) with *catB3* but only until position 442 bp (full-length *catB3* is 633 bp) and there is no difference between *E. coli* and *KpSC*. Therefore, we concluded that *catB4* is a truncated variant of *catB3* and suggest it should be called *catB3Δ443–633*. IS26-mediated deletions of adjacent DNA have previously been reported[25,26].

We traced *catB4* back in the literature and found it was first described, to the best of our knowledge, in two plasmids (pEK499 and pEK516)[27]. Both plasmids had been moved into different *E. coli* lab strains and did not confer CHL resistance, consistent with our data showing that *catB4* is not a functional CAT. Because this had been annotated as a *catB4* variant in the ARG-ANNOT (and SRST2) databases[28] we tested two additional databases for AMR genes. Both, the CARD database[29] and ResFinder[30] called the truncated *catB3* correctly.

We selected five *E. coli* isolates for long-read sequencing to investigate the genomic context of *catB3*, *catB3Δ443–633* and *catA1* (Supplementary Table 2). Aligning four closed contigs containing *catB3* and its truncated variant showed a similar and common genetic feature of *aac(6')-Ib-cr– blaOXA–1–catB3* flanked by two IS26 elements. There was an additional IS26 downstream of the wild-type *catB3* gene (Fig. 3b). In the four isolates, this feature is located on a different replicon, on the chromosome and three different IncF plasmids suggestive of moving as a transposable element independent of a single plasmid-type. As this genetic feature has previously been shown to be part of an integron cassette[31,32], we ran IntegronFinder and found *attC* sites upstream of *blaOXA–1* and downstream of *catB3*, however, the truncation of *catB3* removed the *attC* site, likely precluding its ability to move between or within integron(s). Further, the integrase gene is missing in all but one isolate (CAE137) where it is interrupted by IS26 (Fig. 3b).

**Table 1 | Isolates functionally characterised in this study**

| ID | ST | Species | CHL MIC | R/S | CHL resistance gene(s) |
|---|---|---|---|---|---|
| CAE137 | 648 | *E. coli* | 512 | R | catA1, catB4 |
| CAD110 | 424 | *E. coli* | 512 | R | catA2, catB4, cmlA5 |
| CAB199 | 44 | *E. coli* | 128 | R | catB4, floR |
| CAC122 | 101 | *E. coli* | 32 | R | cat_pC221, cmlA1 |
| CAC10A | 167 | *E. coli* | 16 | R | catA1 |
| CAG10A | 46 | *E. coli* | 16 | R | catB3 |
| CAI1O4 | 46 | *E. coli* | 16 | R | catB3 |
| CAD10B | 46 | *E. coli* | 16 | R | catB3 |
| CAD105 | 46 | *E. coli* | 16 | R | catB3 |
| CAC10H | 617 | *E. coli* | 16 | R | catB4 |
| CAN108 | 617 | *E. coli* | 16 | R | none |
| CAB17W | 131 | *E. coli* | 16 | R | none |
| CAE11U | 167 | *E. coli* | 16 | R | none |
| CAB184 | 44 | *E. coli* | 8 | S | catA1, catB4 |
| CAI10X | 167 | *E. coli* | 8 | S | catA1 |
| CAI10E | 167 | *E. coli* | 8 | S | catA1 |
| CAC124 | 131 | *E. coli* | 8 | S | catB4 |
| CAD13N | 410 | *E. coli* | 8 | S | catB4 |
| CAE12L | 44 | *E. coli* | 8 | S | catB4 |
| CAD13L | 410 | *E. coli* | 8 | S | catB4 |
| CAC116 | 167 | *E. coli* | 8 | S | none |
| CAC10W | 10 | *E. coli* | 8 | S | none |
| CAE12U | 10 | *E. coli* | 8 | S | none |
| CAI10Z | Novel | *E. coli* | 4 | S | catA1, catB4 |
| CAM10K | 443 | *E. coli* | 4 | S | catA1 |
| CAJ10A | 131 | *E. coli* | 4 | S | none |
| CAC13M | 44 | *E. coli* | 4 | S | none |
| CAF11G | 15 | *KpSC* | 512 | R | catA1, catB4, cmlA5 |
| CAC11N | 45 | *KpSC* | >512 | R | catA2, catB4, cmlA5 |
| CAE12L | 340 | *KpSC* | >512 | R | catA2, cmlA5 |
| CAE135 | 340 | *KpSC* | >512 | R | catA2, cmlA5 |
| CAB15M | 152 | *KpSC* | 256 | R | catA1, catB4 |
| CAC13Y | 307 | KpSC | 64 | R | catB4 |
| CAF102 | 15 | *KpSC* | 8 | S | catB4 |
| CAC14G | 1427 | *KpSC* | 8 | S | catB4 |
| CAB16G | 14 | *KpSC* | 4 | S | catB4 |
| CAC105 | 307 | *KpSC* | 4 | S | catB4 |
| CAC13D | 340 | *KpSC* | 4 | S | catB4 |
| CAG19R | 1427 | *KpSC* | 4 | S | catB4 |
| CAD137 | 14 | *KpSC* | 4 | S | none |
| CAL11H | 307 | *KpSC* | 4 | S | none |
| CAD1O7 | 14 | *KpSC* | 2 | S | none |

*ID* name/identifier of isolate, *ST* sequence type, *CHL* chloramphenicol,
*MIC* minimal inhibitory concentration (in μg/mL), *R* resistant, *S* susceptible.

### IS*5* insertion into *catA1* promoter causes reversal of CHL resistance

Of 9/42 isolates with a *catA1*, only four had a CHL resistance phenotype consistent with its genotype (Fig. 2). We, therefore, PCR-probed these nine isolates with primers specific for *catA1* to confirm its presence. Three isolates containing *catA1* that were CHL-R had the expected amplicon size of 150 bp, however, three CHL-S isolates and CAC10A had a larger amplicon than expected of ~1400 bp, (Supplementary Fig. 4). We then assembled the genomes of isolates with *catA1* and aligned the contigs containing *catA1*. Two contigs (from isolates CAI10X and CAI10E) showed a presumed insertion of 1199 bp, starting

6 bp upstream of the CDS, matching the larger than expected PCR amplicon and BLASTn revealed a 100% match to an IS*5* element. In addition, we purified and Sanger-sequenced the ~1400 bp PCR product and confirmed the insert as an IS*5* element in three of the isolates. Long-read sequencing of isolate CAC10A confirmed the insertion of IS*5* as a single element into the promoter region of *catA1* (Fig. 3c). Overall, of the isolates with *catA1*, 3/9 were CHL-R with detectable CAT activity (Fig. 2) and had an expected *catA1* amplicon size (150 bp); isolate CAC10A was CHL-R with an MIC (16 mg/L) close to the breakpoint, no detectable CAT activity and IS*5* inserted into the *catA1* promoter region. The remainder (5/9) were CHL-S, without detectable activity on the rCAT assay; this was likely mediated by IS*5* insertion interfering with transcription.

We adjusted primers from a previously developed high-resolution melting (HRM) assay[33] to capture IS*5*-*catA1* and the truncated *catB3Δ*[443–633], and demonstrated that they can be clearly distinguished from the wild-type genes (Supplementary Fig. 5), highlighting the potential of this molecular diagnostic test for CHL resistance to discriminate between functional and non-functional *cat* genes.

### Degradation of *cat* genes by insertion sequences is stable
IS*5* insertion into the promoter of *catA1* and truncation of *catB3* are the main mechanisms for the observed re-emerging CHL susceptibility in our collection of isolates. To test if these mutation and insertion events could be reversed and potentially result in a rapid re-emergence of *cat* expression and CHL resistance, we experimentally evolved three CHL susceptible isolates: CAI10Z (*catA1*, *catB3Δ*[443–633]), CAM10K (*catA1*) and CAI10X (*catA1*) in LB broth with increasing concentrations of CHL for 7 days (*n* = 3 per isolate). An equal number of replicates were evolved in LB without selection. All evolved populations under CHL selection grew until 64 μg/mL CHL (8–16-fold increase in MIC). Increasing incubation for another 24 h enabled recovery of some populations in 128 μg/mL (i.e. clearly visible growth) but none of the CHL populations grew in 256 μg/mL CHL (Supplementary Fig. 6a). All control populations evolved in LB grew until the experiment was stopped at day 7. Since we were interested if CHL pressure can result in a re-activation of *cat* genes, we tested all evolved populations with the rCAT assay (LB evolved from day 7, CHL evolved last surviving population, day 5, day 6 or day 7). All populations examined tested negative on the rCAT assay (Supplementary Fig. 6b) and PCR-probing of evolved strains showed that the IS*5* insertion was still present upstream of *catA1* (Supplementary Fig. 6c), confirming that other mutations rather than (re-) expression of *catA1* are responsible for the resistance phenotype.

### The genomic locus of *catB3Δ*[443–633] is conserved and widespread
The truncated *catB3Δ*[443–633] is common among our isolates and seems to be highly conserved in the genomic context of *aac(6')-Ib-cr*–*bla*[OXA–1]–*catB3*. To investigate potential drivers, we ran a co-occurrence analysis across 772 genomes in our collection from Malawi (68 of the 840 total isolates failed assembly quality check) (Supplementary Fig. 7 and Supplementary Data 2). We applied the probabilistic model from Veech et al.[34] to determine whether each pair of genes had an observed co-occurrence which was significantly different from the expected co-occurrence. The co-occurrence relationships across all AMR genes with at least one other gene are depicted in Supplementary Fig. 7. Both, *catB3* and *catB3Δ*[443–633] had a positive co-occurrence relationship with *bla*[OXA–1] and *aac(6')-Ib-cr* and the beta-lactamase genes *bla*[CTX-M-15] and *bla*[TEM-1B]. The *catB3* and *catB3Δ*[443–633] genes had a negative co-occurrence as they were never found to co-occur on the same genome (Fig. 4).

In our isolates, *catB3Δ*[443–633] is much more common than *catB3*, and thus we investigated if this pseudogene is restricted to Malawi or common elsewhere. We queried "catB3" in NCBI's Microbial Browser for Identification of Genetic and Genomic Elements (Micro-BIGG-E)[35] and investigated 46,667 isolates (data download: 4/8/2023) with an

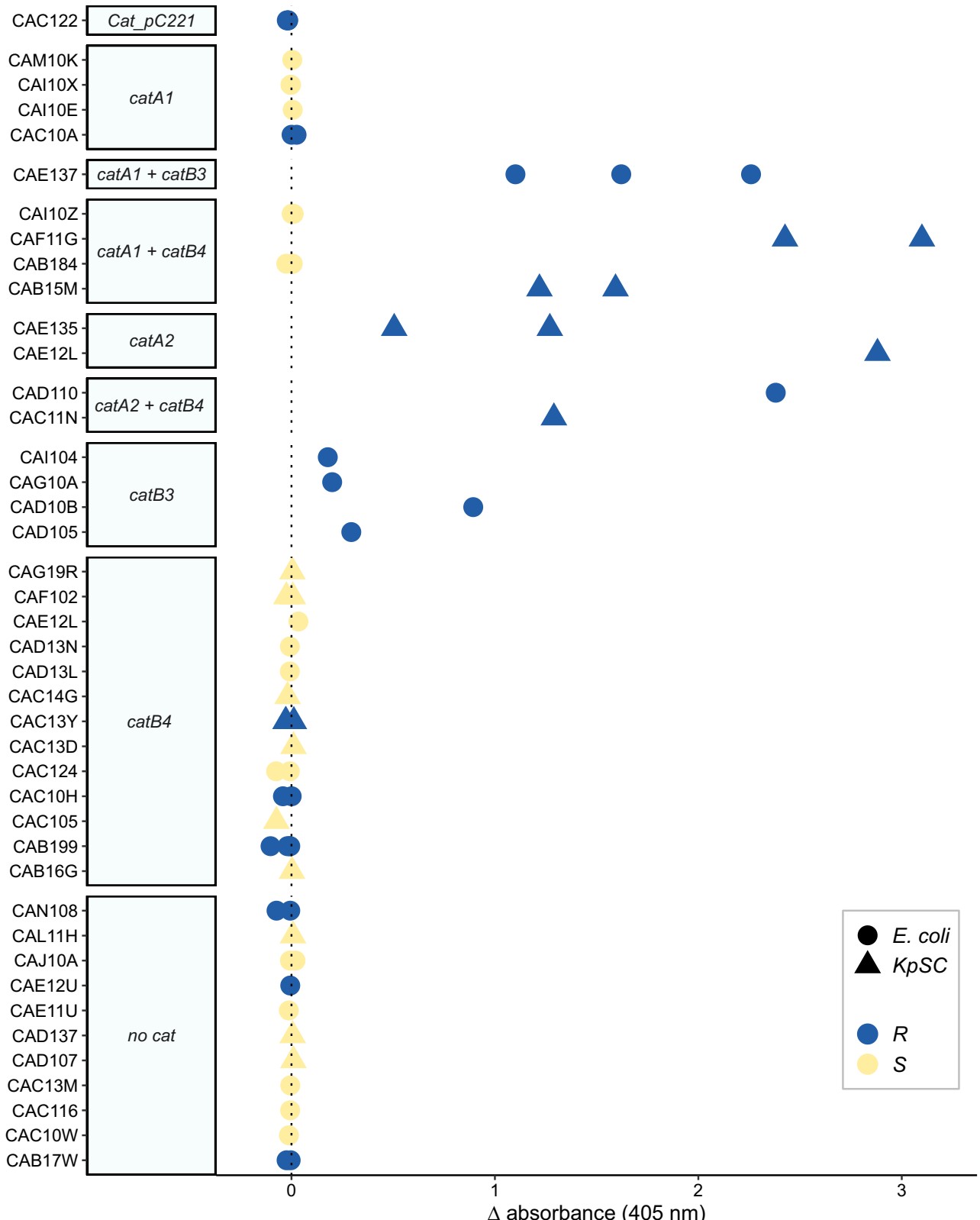

**Fig. 2 | CAT enzyme activity.** CAT enzyme activity was measured using the rCAT assay. The difference ((signal + CHL) − (signal − CHL) for a single isolate (*n* = 2–6)) in absorbance at 405 nm is given for each of the 42 isolates. The colour indicates if the isolate is phenotypically resistant (R, blue) or susceptible (S, yellow) to CHL based on broth microdilution and the shape indicates *E. coli* (circle) or *K. pneumoniae* (triangle). The *Y*-axis is ordered according to the presence of *cat* genes. Source data are provided as a Source Data file.

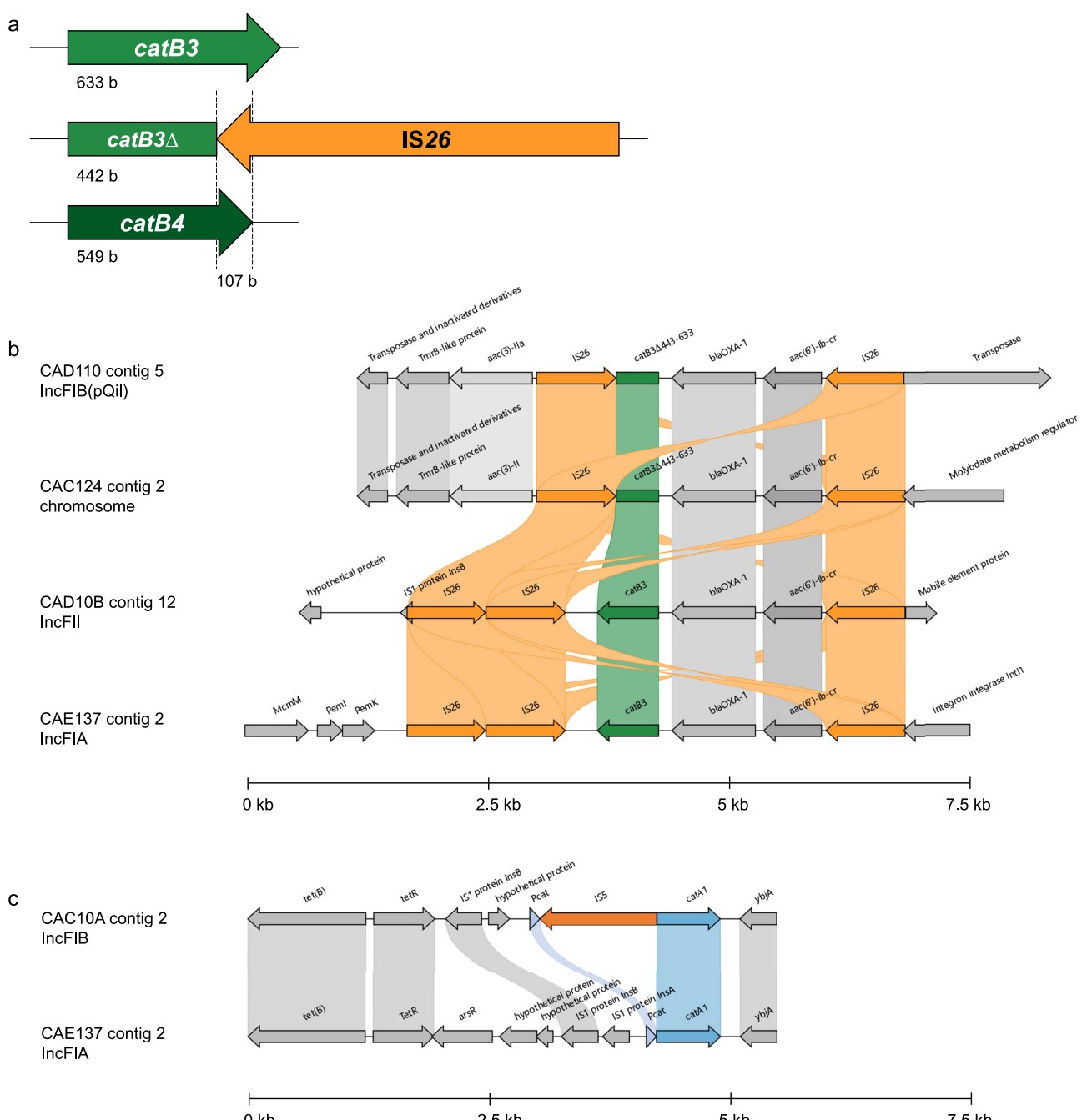

**Fig. 3 | *Cat* gene degradation by insertion sequences. a** Schematic of IS*26* truncation of *catB3*. **b** Alignment of four contigs from four different isolates containing either *catB3* or *catB3Δ⁴⁴³⁻⁶³³*. **c** Alignment of two genomes containing *catA1*, with and without an IS*5* insertion into the *catA1* promoter.

annotated *catB3* gene. Strikingly, 30,819 (66%) showed a coverage to the reference of 70%, which corresponds to the IS*26* truncated variant described here (Supplementary Fig. 8a). We tried to trace back the emergence of this truncation. From 2006 onwards (only including years with >100 isolates) the truncated variant was dominant over the wild type, and we could not conclude when it emerged (Supplementary Fig. 8b). *Klebsiella* spp., *E. coli*, and *Enterobacter* spp. dominantly show the truncated gene whereas *Salmonella*, *Acinetobacter*, and *Pseudomonas*, typically harbour wild-type *catB3* (Supplementary Fig. 8c). Differentiating by host where the isolates have been collected is heavily biased towards humans where the dominant truncated variant is much more common. This is also seen in companion animals (cats and dogs) whereas, in most food animals, the wild-type proportion is close to 100%, perhaps

indicative of ongoing selection for CHL resistance through the use of phenicols in veterinary medicine and agriculture (Supplementary Fig. 8d). Most geographic locations show higher proportions of the truncated variant with a few exceptions in China and Australia (Supplementary Fig. 8e).

Next, we investigated the spread of *catB3Δ⁴⁴³⁻⁶³³* in the context of STs to potentially link clonal expansion and *catB3Δ⁴⁴³⁻⁶³³* carried by these clones *in E. coli*. The *catB3Δ⁴⁴³⁻⁶³³* gene was restricted to six STs (410, 44, 648, 405, 617, and 131) in our Malawi isolates (Fig. 5), and five of those STs are among the top eight most common STs in our collection (Supplementary Fig. 9). We expanded our analysis and leveraged an existing collection of 10k genomes consisting of the top 100 common STs in *E. coli* (previously curated and downloaded from Enterobase (2020)[36] where each ST was randomly sampled to select

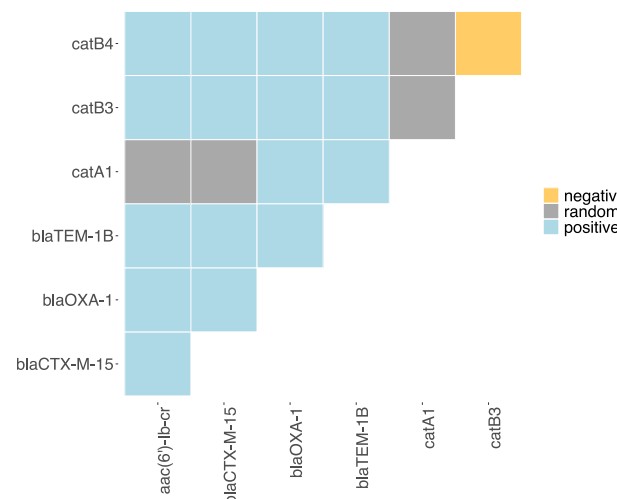

**Fig. 4 | Co-occurrence network of selected AMR genes.** Heatmap displaying co-occurrence relationships between AMR genes as either positive (blue), random (grey) or negative (orange). These are probabilistic values based on the difference in expected and observed frequencies of co-occurrence between each pair of genes, these values were obtained by applying the probabilistic model from[34]. Co-occurrence across select genes including *cat genes*, *aac(6')-Ib-cr*, *bla*OXA-1, *bla*CTX-M-15, and *bla*TEM-1. Source data are provided as a Source Data file.

100 genomes. All six STs with *catB3Δ443−633* in Malawi also dominantly harbour this *cat* gene in this extended genome collection (Fig. 5). In phylogroup B2, which is predominantly human isolates associated with urinary tract and bloodstream infections, *catB3Δ443−633* is common in ST131 and to a lesser extend ST1193 (Fig. 5), both of which represent recently emerged globally dominant antibiotic resistant clones[37,38]. The ubiquitousness of *catB3Δ443−633* may thus be linked to its association with distinct and successful lineages.

## Discussion

In Malawi, as in much of sub-Saharan Africa, the rapid spread of ESBL-E coupled with the scarcity of Watch and Reserve antibiotics (i.e. carbapenems) has rendered many severe bacterial infections untreatable, therefore the re-emergence of CHL susceptibility is potentially important. 34.2% (1666/4874) of non-salmonella Enterobacterales were phenotypically CHL susceptible in bloodstream infections in Malawi from 1998–2016[5]. Here, we analysed a collection of phenotypically and genotypically characterised *E. coli* and *KpSC* from Malawi to understand the molecular basis of CHL-S genotype–phenotype mismatches.

Since many phenotypically susceptible isolates in our collection carried *cat* genes, we first investigated if they were still functional and expressed. None of the CHL-S isolates we tested had functional CATs and we found that *cat* gene interruption was caused by insertion sequences. This is in contrast to a previous study reporting an *E. coli* strain being susceptible to CHL despite a functional CAT[39] which was later attributed to a low level of acetyl-coenzyme A linked to mutations in efflux pumps[40]. Combined functional analysis of *catB4* with CAT enzyme assays, expression in a clean genetic background, as well as genomic investigation confirmed that *catB4* is non-functional and is in fact a *catB3* that has been truncated by an IS*26* element. In other cases, an IS*5* element has integrated into the promoter region of *catA1* and likely interfered with transcription. In the latter case, calling AMR genes from all databases with sequence data will yield a wild-type gene and currently classify the isolate as resistant since the mutation is outside of the CDS. The truncated *catB3* will be called correctly if the annotation of *catB4* is removed in the ARG-ANNOT/SRST2 databases, which we strongly support.

To effectively re-introduce CHL as a reserve treatment option for Enterobacterales infections confirmed as 3GC-resistant, we must consider the potential for reversion to CHL resistance. Isolates carrying IS*5-catA1* could potentially rapidly revert to high-level resistance since the *catA1* CDS is still present in the genome. Our data using experimental evolution suggest that this is not the case and selection with CHL did not lead to the expression of *catA1*. One of the evolved isolates co-carried a truncated *catB3* and we expected no reversion to a functional *catB3* upon CHL selection because the missing 3' prime end of the gene is no longer present in the genome. Indeed, no functional *cat* emerged. Resistance to CHL will ultimately re-emerge when CHL is used more frequently but recent efforts of antibiotic stewardship in Malawi aim to keep selection pressures low. Critically, CHL was once the empirical intravenous agent for the management of sepsis across much of Africa and was available in the community, we are proposing limited re-introduction for blood culture-confirmed infection, which results in orders of magnitude less use.

A decrease in CHL-resistant isolates has also been reported for *Salmonella* Typhimurium, which was caused by the loss of CHL resistance genes by lineage replacement[41]. Our study adds to a developing evidence base for CHL in the treatment of MDR Gram-negative pathogens; several studies have reported high or increasing rates of phenotypic CHL susceptibility among MDR Gram-negative bacteria suggesting CHL as a viable alternative treatment option in those settings[42–44]. Further, CHL is an affordable and useful antimicrobial in terms of bioavailability, tissue penetration and broad spectrum of action[45,46]. However, the prevalence of CHL susceptibility and the rare but severe side-effects of CHL, preclude the use of CHL for empirical management of sepsis in our setting: we instead envisage CHL to be used in critically ill adults (CHL is contraindicated for children given its toxicity) and hospitalised patients with confirmed ESBL-E and CHL-S infection as a reserve agent when no treatment alternative is available. This has the added advantage of keeping selection pressure for CHL resistance low but does require rapid determination of CHL susceptibility phenotype.

In the Malawian isolates the gold-standard AST correctly determined phenotypic CHL susceptibility, however, rapid molecular diagnostics have the potential to be faster and low cost, but require knowledge of resistance mechanisms. We applied the HRM assay, previously developed to detect acquired CHL resistance genes[33], to distinguish between functional and non-functional *cat* genes found in our study.

Co-occurrence analysis showed that *catB3* and *catB3Δ443−633* nearly always co-occurred with *aac(6')-Ib-cr* and *bla*OXA–1. This conserved feature of *aac(6')-Ib-cr-bla*OXA–1*-catB* flanked by IS*26*, which was confirmed by long-read sequencing, has previously been associated with an integron[32] and was found to contain *attC* sites when *catB3* was present. However, the truncation by IS*26* removed one of the *attC* sites. This, in combination with the loss of the integrase gene, likely led to functional loss of mobilisation of genes within the integron and the high conservation of this locus is confirmatory. The association of *catB3Δ443−633* with *bla*CTX-M-15, the most common ESBL gene found in *E. coli* isolates worldwide[47] and among our Malawian isolates[17,18], may point towards the co-location of these genes on a plasmid, though a larger selection of isolates will need to be (long-read) sequenced to determine the spread of such a plasmid. Selection for 3GC resistance and co-occurring *bla*CTX-M-15 with *catB3Δ443−633* could have led to a genetic hitchhiking of the latter which could explain the high levels of CHL susceptible isolates among ESBLs. However, our isolate collection is heavily biased towards ESBL and may thus limit the interpretation of this co-occurrence. Another limitation of our co-occurrence analysis is the limited set of (resistance) genes and isolates, which will not account for all genetic interactions influencing CHL resistance.

Expanding our analysis of *catB3Δ443−633* revealed that this single truncation is globally more common than the wild-type gene. It is

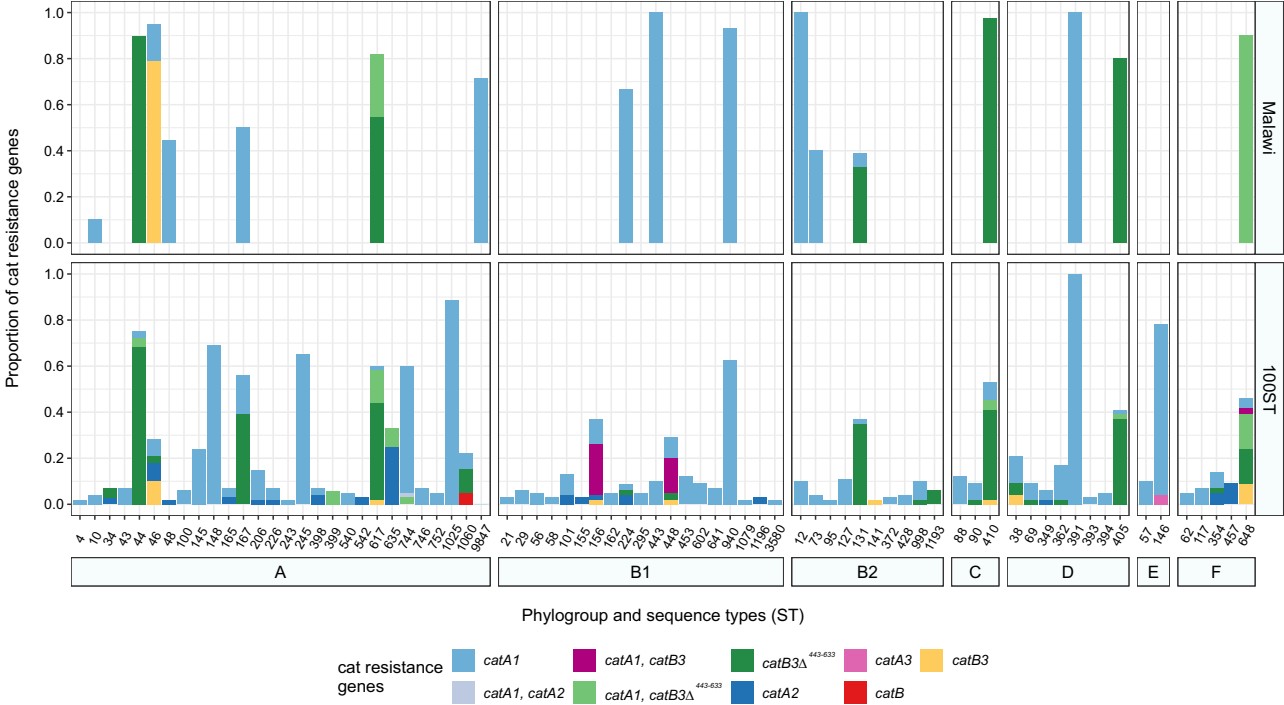

**Fig. 5 | *Cat* gene occurrence and proportions per ST.** *E. coli* isolates from Malawi included in our study (top panel) and in a 10k genome collection of 100 randomly selected genomes of the top 100 most common STs from *E. coli* (bottom panel). Source data are provided as a Source Data file.

intriguing that a truncated non-functional gene is so widespread, and we hypothesise this is due to the specific genetic context of a loss of integron activity, and association with other AMR genes, as well as with IS*26*. It has previously been shown that IS*26* and pseudo-compound transposons (i.e. IS*26* bounded transposons) move 50 times more frequently if there is another IS*26* to target[48] and they move by co-integration[49]. This mode of mobility likely enhances persistence. Additionally, an association of *catB3Δ*[443–633] with globally dominant lineages of *E. coli* (e.g. ST131) as seen in our ST analysis, could be a likely driver that has contributed to the widespread occurrence of this truncated variant.

Our study highlights that antibiotic susceptibility can re-emerge following a reduction or cessation of use. Plasticity and the high levels of horizontal gene transfer among Enterobacterales can result in the degradation of AMR genes by mobile genetic elements which can persist in the absence of selection. Further, the context of the AMR gene determines the phenotypic resistance and database curation is crucial to infer resistance from the genotypes. We strongly recommend integrating functional context into databases and including additional features such as genetic context, promoters and transcription start sites. Many AMR genes only cause clinically relevant resistance when they are moved to locations in the genome where expression is upregulated, such as downstream of a strong promoter[50]. This is problematic with functional metagenomic discovery of AMR genes since the mere presence of an AMR gene is not indicative of phenotypic resistance. Lastly, our data support the reintroduction of CHL, carefully balancing its potentially severe side effects, as a last-line treatment option for patients critically ill with ESBL-E infections in Malawi and similar settings.

## Methods

### Bacterial isolates and genomes

Bacterial isolates and their genomes investigated in our study are 566 *E. coli* and 274 *KpSC* isolates that had previously been isolated at QECH in Blantyre, Malawi as part of two different studies. 164 (93 *E. coli*, 71 *KpSC*) isolates were from a sentinel surveillance study of

bacteraemia and isolated between 1998 and 2016[5]. 676 isolates (473 *E. coli*, 203 *KpSC*) were from a study investigating the ESBL-Enterobacterales colonisation in the gut of Malawian adults and collected in 2017 and 2018 (DASSIM study[14]). Isolates were sequenced on the Illumina HiSeq X10 (paired-end 150 bp) for the colonisation study[14] and on the Illumina HiSeq 2000 (paired-end 100 b) for the bacteraemia study[15,16].

Our study complies with ethical regulations for re-use of previously collected bacterial isolates. The bacteraemia study was approved by the College of Medicine Research Ethics Committee (COMREC) of the University of Malawi (P.08/14/1614). The DASSIM study was approved by the Malawi College of Medicine (P.11/16/2063) and the research ethics committees of the Liverpool School of Tropical Medicine (16–062).

### Antimicrobial susceptibility testing

AST was performed using broth microdilution according to the EUCAST guidelines (v.4.0) in cation-adjusted MullerHinton broth (MH2, 90922, Merck) in duplicates. If results were inconsistent or different to the CHL susceptibility phenotype previously determined with disc diffusion, AST was repeated. Broth microdilution data was used to determine the phenotypic classification of the subset of 42 isolates. Isolates are classified according to EUCAST (v.12.0); isolates with a MIC ≤ 8 μg/mL were classified as susceptible and isolates > 8 μg/mL as resistant. In EUCAST v.13.0, CHL breakpoints are no longer listed, stating a screening cut-off (CHL MIC > 16 μg/mL) can be used to distinguish wild type from acquired resistance.

### Functional CAT assay (rCAT)

We adjusted the rCAT assay[19] to enable read-out in 96-well plates using a spectrophotometer. Cultures of isolates to be tested were set up in replicates in 1–2 mL LB broth and incubated at 37 °C with 220 rpm. 100 μL of overnight culture was transferred into a microcentrifuge tube containing 500 μL PBS and centrifuged at 6000×*g* for 3 min to wash and pellet cells. The supernatant was completely removed using a pipette, the pellet resuspended in 250 μL lysis buffer (1 M NaCl, 0.01 M

EDTA, 0.05% SDS) and cells lysed while incubating for 1 h at 37 °C with occasional vortexing the tubes. Next, 50 μL of the lysed cells were transferred into each of four wells of a 96-well plate. To each well 50 μL reaction buffer was added (for 1 mL, 400 μL 0.2 M Tris; 400 μL 5 mM acetyl-CoA; 200 μL 10 mM DTNB were mixed). DTNB was added last to the wells, since DTNB was reported to inhibit certain CAT enzymes[19]. DTNB was always prepared fresh since stored DTNB (for even 1 day at 4 °C in the dark) increased the background. Acetyl-CoA was always prepared on ice and aliquots immediately frozen at −20 °C and used within two weeks.

50 μL 5 mM CHL was added to the first two wells and 50 μL ddH₂O to wells three and four of each isolate. The plate was incubated for >10 min at 37 °C−longer incubation led to a higher signal for positive samples while the background did not change−up to 1 h tested.

The plate was read with a plate reader (GloMax® Discover Microplate Reader (Promega) or CLARIOstar Plus Microplate Reader (BMG Labtech)) at an absorbance of 405 nm or 412 nm and absorption from lysed cells without CHL was subtracted from the value with CHL, hence, positive values indicate CAT activity, and no activity gives a value around zero.

## Competent cells and transformation

MG1655 was made chemically competent as follows: A single colony of MG1655 was picked from an LB agar plate and incubated in 3 mL LB broth at 37 °C with 220 rpm for 16 h. The culture was diluted 1:100 (1 mL in 100 mL) in a 500 mL flask and incubated at 37 °C with 220 rpm until OD$_{600}$ reached 0.3−0.4. The flask was immediately cooled in an ice-water slurry before the cells were transferred to pre-chilled 50 mL Falcon tubes. Cells were centrifuged at 3500×g at 4 °C, the supernatant discarded and the cell pellet resuspended in 10 mL ice-cold CaCl₂ solution (60 mM CaCl₂, 15% glycerol, 10 mM PIPES pH 7). The centrifugation step was repeated, and resuspension was kept on ice for 30 min. After another centrifugation step the cells were resuspended in 1 mL ice-cold CaCl₂ solution and aliquots of 50 μL or 100 μL were frozen at −70 °C.

For transformation, the chemically competent cells were thawed on ice. Two microlitres of plasmid DNA (pEB1-variants; 0.5 ng/mL) were added and incubated on ice for 30 min. Cells were heat-shocked for 30 s at 42 °C in a water bath and immediately put back on ice for 2 min. 900 μL SOC media was added to cells and incubated for 1 h at 37 °C 220 rpm before being spread on LB kanamycin (50 μg/mL) plates.

## Cloning and functional expression of *cat* genes

*Cat* gene variants were amplified using primers spanning the coding sequence with 20 bp overhangs homologous to the pEB1-plasmid cloning site (Supplementary Table 1). Gibson assembly[51] (NEBuilder® HiFi DNA Assembly Master Mix, E2621S, NEB, UK) was used to clone *catA1*, *catA2*, *catB3*, and *catB4* into pEB1. Plasmids were verified by colony PCR and Sanger sequencing of the insert regions using pEB1-sequencing primers (Supplementary Table 1). Plasmids were extracted using QIAprep Spin Miniprep Kit (27106, Qiagen) and transformed into competent MG1655.

## Experimental evolution of CHL resistance

Isolates were retrieved from frozen stock on LB agar plates. A single colony was used to inoculate a 1 mL starting culture and incubated at 37 °C with 220 rpm (the ancestor). Then, for each tested isolate the starting culture was diluted 1:100 into three tubes with 3 mL LB broth (control populations) and three tubes with LB and 0.5× the MIC of CHL (selected populations) of the isolate. All populations were incubated for 24 h at 37 °C with 220 rpm and then again diluted 1:100 in fresh LB broth with or without CHL. In the selected population the CHL concentration was doubled with each passage. This was repeated for 7 days or until the population became extinct.

## DNA extraction

DNA was extracted with MasterPure™ Complete DNA and RNA Purification Kit (MC89010, LGC Biosearch Technologies). Isolates were grown overnight on blood agar (PP0120-9090, E&O labs) at 37 °C and colonies were picked and washed in 500 μL of sterile PBS (10010023, Fisher Scientific) by centrifugation at 10,000×g for 3 min. Pellets were resuspended in 300 μL Tissue and Cell Lysis Solution supplemented with 1 μL of Proteinase K solution. Tubes were incubated at 65 °C shaking at 1000 rpm for 5 min. Samples were placed on ice for 3 min. 1 μL of RNase A solution was added and the samples were incubated at 37 °C for 30 min.

Samples were placed on ice for 5 min then 150 μL of MPC Protein Precipitation Reagent was added and samples vortexed for 10 s. The resulting protein debris was pelleted by centrifugation at 10,000×g for 10 min at 4 °C. The supernatant was aspirated and added to 500 μL of isopropanol (15631700, Fisher Scientific) and the tube was inverted 30−40 times to precipitate the nucleic acids which were then pelleted by centrifugation at 10,000×g for 10 min at 4 °C and supernatant discarded. The pellet was washed twice in 70% ethanol and resuspended in 50 μL of TE Buffer. The quantity of DNA was then measured using Qubit dsDNA Quantitation, Broad Range kit (Q32850, Thermo Scientific) on a Qubit 4 Fluorometer (Thermo Scientific).

## Long-read sequencing

Five *E. coli* from the subset of 42 isolates with a genotype−phenotype mismatch were re-streaked from glycerol stocks onto antibiotic-supplemented LB agar and incubated at 37 °C for 16 h. A single colony was picked and transferred to LB broth supplemented with antibiotics. Antibiotic supplementation included CHL and/or ampicillin depending on the resistance profile of each strain. Genomic DNA was extracted using the Firemonkey High Molecular DNA Extraction Kit (Revolugen, UK) according to the manufacturer's protocol.

The strains were long-read sequenced on a MinION device (Oxford Nanopore Technologies (ONT), UK). Library prep was carried out according to the manufacturer's protocol (ONT, UK) using the SQK-NBD114.24 Ligation Sequencing and Native Barcoding Kit. The DNA library was quantified at several stages in the library prep using a Qubit Fluorometer (ThermoFisher Scientific, Massachusetts, USA), the TapeStation 4100 (Agilent, California, USA) was used to determine the molar concentration and 10−20 fmol was loaded onto the flow cell. Sequencing was carried out using an FLO-MIN114 (R10.4.1) flow cell (ONT, UK) on a MinION Mk1B sequencer, running for 72 h at a translocation speed of 400 bp/s. Data acquisition used the MinKNOW software (v22.08.9).

## Bioinformatic analysis

AMR genes from short-read data were called using ARIBA v.2.14.6[52] and the curated ARG-ANNOT database used by SRST2[28]. For the long-read data, the raw fast5 files from the MinION sequencing run were basecalled using Guppy v6.4.2 with the super accuracy (sup) model for DNA sequencing on the R10.4.1 flowcell with the E8.2 motor protein and the 400 bp/s translocation speed. Nanoplot v1.38.1[53] was used to check the quality of the sequencing reads and the parameters of the sequencing run. The basecalled fastq files were demultiplexed using the guppy barcoder from Guppy v6.3.8 with the SQK-NBD114-24 kit. The sequence reads were assembled with flye v2.9.1[54] using trestle mode. Seqkit v0.15.0[55] was used to determine basic statistics and contig sizes (Supplementary Table 2). The assembled contigs were annotated with RAST[56] and any query MGEs aligned against the ISfinder database[57], Mobile Element Finder database[58] and IntegronFinder[59]. Resfinder was used to determine resistance genes[60]. Annotated assemblies were visualised in Snapgene, and clinker[61] was used on the Genbank files to align the contigs and highlight any homologous genetic clusters.

## Co-occurrence analysis of AMR genes

Genome sequences (all Illumina short-read data) for our assembled isolate collection were obtained from the European Nucleotide Archive: PRJEB8265, PRJEB28522, PRJEB26677, and PRJEB36486[15–18]. Fastq files were downloaded in April 2023. Initially, the paired-end short-read fastq files were downloaded and trimmed with cutadapt[62]. All reads were analysed with FASTQC and found to pass over half the quality control determinants, sequences with a Phred quality score less than Q20 across the length of the reads were excluded. Reads were assembled using SPAdes v3.11.1[63]. 68 genomes failed initial QC or assembly and the dataset taken forward for analysis contained 495 *E. coli* and 277 *KpSC* genomes. The quality of the assembled contigs was assessed using the stat command from Seqkit v0.15.0[55]. The final dataset contained 772 assembled fasta files.

Abricate v0.0.9 (https://github.com/tseemann/abricate) was used to screen for AMR genes and create output tables, sequences were compared against the Resfinder[60] database at 60% minimum length and 90% percentage identity using the BLASTn algorithm. The R programming language v 4.3.1 was used to convert the Abricate output tables into an appropriate format for further data analysis and visualisation, including a binary presence/absence AMR gene table, with the tidyverse (v1.3.0) package[64]. Any annotation of *catB3* with less than 75% coverage was labelled *catB4*. The cooccur package v1.3[65] was used to create a co-occurrence matrix containing the probabilistic values which represent whether a co-occurrence relationship is observed significantly more or less than could have happened by chance. This used the probabilistic model of co-occurrence[34]. If a pair of genes were observed to co-occur significantly less than expected by chance their relationship was termed negative, if they were observed to co-occur significantly more it was termed positive and any pair of genes which were observed to co-occur with no significant difference from their expected value were termed random. A significant difference was defined as having a *p*-value ≤ 0.05. Select genes conferring resistance to the phenicol, beta-lactam and aminoglycoside drug classes were then selected from the original presence/absence table for further analysis. Heatmaps to display co-occurrence were visualised using ggplot v 3.4.3[64] and pheatmap v 1.0.12. Clustering in the heatmap used hierarchical clustering with Euclidian distance and the complete method.

## ST analysis

The genome assemblies of 100 randomly selected isolates from each of the 100 most common STs in Enterobase were downloaded on 18/12/2020. Extra genomes from ST391, ST44, ST940, and ST9847 with a release date before 18/12/2020 were added later to match with the STs from the Malawi dataset. The STs of the Malawi dataset were determined using MLST (v2.23.0). The presence of *cat* genes was detected using AMRFinderPlus version 3.11.20 with database version 2023-09-26.1[66]. The data was analysed using R (v4.1.3) with the tidyverse package (v1.3.1) and plotted using ggplot2 (v3.4.2), ggpubr (v0.4.0), and RColorBrewer (v1.1–2).

## Statistics and reproducibility

Data were analysed using R. No statistical method was used to predetermine sample size. No data were excluded from the analyses. The experiments were not randomized. The investigators were not blinded to allocation during experiments and outcome assessment.

## Reporting summary

Further information on research design is available in the Nature Portfolio Reporting Summary linked to this article.

## Data availability

Supplementary Figs. 1–9 and supplementary Tables 1 and 2 are accessible in the supplementary information of this manuscript. Reads from all isolates previously sequenced and used in this study are accessible in the European Nucleotide Archive (ENA) under project IDs PRJEB8265, PRJEB26677, PRJEB28522, and PRJEB36486 and ENA accession numbers for isolates linked to metadata are available in the Supplementary Data 1. Long-read sequence data have been submitted to the National Center for Biotechnology Information (NCBI) under the BioProject ID PRJNA1040831, BioSample accession numbers for sequenced isolates are in Supplementary Table 2. Source data are also accessible from the GitHub repository https://github.com/FEGraf/CHL-Malawi, https://zenodo.org/records/13861919[67]. Source data are provided with this paper.

## Code availability

The R scripts used to generate analyses, figures, and tables are available from the GitHub repository https://github.com/FEGraf/CHL-Malawi, https://zenodo.org/records/13861919[67].

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

## Acknowledgements

This work was supported by iiCON (infection innovation consortium) via UK Research and Innovation (107136) and Unilever (MA-2021-00523N). R.N.G. has been supported by the Medical Research Council via the LSTM-Lancaster doctoral training partnership (grant no. MR/N013514/1). R.N.G and A.P.R. are supported by the Medical Research Council (MRC), Biotechnology and Biological Sciences Research Council (BBSRC) and Natural Environmental Research Council (NERC) which are all Councils of UK Research and Innovation (grant no. MR/W030578/1) under the umbrella of the JPIAMR—Joint Programming Initiative on Antimicrobial Resistance. The funders had no role in study design, data collection and analysis, decision to publish, or preparation of the manuscript. We wish to thank Dr Anne Farewell for *E. coli* MG1655. pEB1-sfGFP was a kind gift from Philippe Cluzel (Addgene: http://n2t.net/addgene:103983). We also wish to thank Dr Simon Wagstaff for the GPU base-calling of Oxford Nanopore data and Dr Nadja Wipf for critically reviewing the R-code.

## Author contributions

The study was conceived by F.E.G., A.P.R., and N.A.F. The methodology was devised by F.E.G., R.N.G., M.D.P., M.A.S., A.P.R., T.E., J.M.L., and N.A.F. Investigations were undertaken by F.E.G., R.N.G., S.G., S.F., E.P.B., A.J.F., M.D.P., M.M., A.T.M.H., P.M., T.E., and J.M.L. Formal analysis was done F.E.G., R.N.G., M.D.P., P.M., M.A.S., A.P.R., T.E., J.M.L., and N.A.F. The original draft was prepared by F.E.G., R.N.G., J.M.L., and N.A.F., and then reviewed and edited by all authors. Supervision was by F.E.G., T.E., A.P.R., and N.A.F.

## Competing interests

The authors declare no competing interests.
