## [Peer Review File · Nature Communications]

REVIEWER COMMENTS

Reviewer #1 (Remarks to the Author):

The manuscript investigate the genetic basis of chloramphenicol (CHL) resistance in bacterial isolates, specifically *E. coli* and *Klebsiella pneumoniae*. It details the identification and functional analysis of chloramphenicol acetyltransferase (CAT) genes, which are responsible for CHL resistance in these bacteria. The study includes a comprehensive examination of various CAT gene variants, including *catA1*, *catA2*, *catB3*, and *catB4*, across different bacterial isolates.

The study reports on the presence of different CAT gene variants in *E. coli* and *Kp* isolates and their association with CHL resistance. Notably, it was found that the functional expression of *catB4* does not confer CHL resistance, suggesting that *catB4* is a non-functional variant. Further analysis revealed that *catB4* is a truncated version of *catB3*, with a significant portion of its sequence matching the IS26 transposase family. This truncation results in the loss of functional CAT activity, leading to the suggestion that *catB4* should be renamed to *catB3Δ443-633*. The study also explores the genomic context of the *catB3Δ443-633* variant, finding it commonly flanked by IS26 elements and part of an integron cassette. This suggests a mechanism for its spread and conservation across different bacterial isolates. The research identifies that the insertion of an IS5 element into the promoter region of *catA1* can reverse CHL resistance, highlighting the complex interplay between genetic elements in determining antibiotic resistance phenotypes.

A co-occurrence analysis among the isolates showed a positive relationship between *catB3Δ443-633* and other antibiotic resistance genes, indicating potential co-selection mechanisms. The analysis also revealed that *catB3* and *catB3Δ443-633* never co-occur in the same genome, suggesting a replacement or competitive exclusion relationship.

Prevalence and Historical Emergence: The study investigates the prevalence and historical emergence of the *catB3Δ443-633* variant, finding it to be widespread and conserved, particularly in human isolates. This prevalence is contrasted with the wild type *catB3*, which is more common in non-human isolates, suggesting different selective pressures in different environments.

The methods used to answer the research questions including disc diffusion and microbroth dilution, long-read sequencing, and functional assays to assess CAT activity and CHL resistance allowed for a detailed genetic and phenotypic characterization of the bacterial isolates.

Overall, the study provides significant insights into the genetic mechanisms underlying CHL resistance in *E. coli* and Kp, highlighting the role of CAT gene variants and other genetic elements in determining resistance phenotypes. The findings have implications for understanding antibiotic resistance mechanisms and could inform strategies for managing and mitigating resistance in bacterial pathogens.

However, I have minor concerns at several aspects:

1. While the study highlighted the conservation and widespread occurrence of the truncated *catB3* Δ 443-633 variant, it focused on a specific genetic context associated with other antimicrobial resistance (AMR) genes and IS26 elements. This limited scope may not fully capture the broader genetic landscape influencing CHL resistance mechanisms like overexpression of multidrug resistance efflux pumps (eg. AcrAB-TolC efflux).
2. The co-occurrence analysis revealed positive relationships between *catB3* Δ 443-633 and certain AMR genes but a negative relationship between *catB3* and *catB3* Δ 443-633, indicating they do not co-occur in the same genome. However, this analysis may not account for all potential genetic interactions influencing CHL resistance.
3. The study emphasized the importance of integrating functional context into databases for accurate inference of resistance from genotypes. It suggested improvements in database curation to consider genetic contexts like promoters and transcription start sites, which are crucial for determining phenotypic resistance accurately. While this highlight is relevant, the same can also be applied to reference guide used to interpret phenotypic disc diffusion or MIC results. I believe this paper, is the best example and place to call for either a globally standardized breakpoint reference guidelines or specific breakpoints curated based on strains circulating at the nation, sub-regional, regional or African level.

Reviewer #2 (Remarks to the Author):

Thanks for the opportunity to review this very interesting paper. The authors compared phenotypic and genotypic markers of chloramphenicol resistance in E coli and Klebsiella, to understand discrepancies between the two susceptibility results. The findings are interesting both from a “basic science” perspective but also in better understanding some of the limitations of molecular resistance testing. The manuscript is well written although there are a few typos here and there and a careful read through is advised.

I have some comments and suggestions:

There is no statement regarding approval by an ethics review board, or of any form of waiver of ethics review.

I think there should be more detail on the isolate collection – when were they collected (so how representative are they of current isolates). Were all the isolates ESBL-E? I thought suppl table 1 (Line 67) would contain details of isolates, but suppl table 1 appears to be a list of scripts. I do think some detail about the isolate collection would be useful.

I also found the methods slightly confusing as they make no reference to to the isolates – all the isolate information is in the results. Given the structure of the manuscript (methods at the end) this is probably fine, but consider a small amount of information about the isolate collection in the methods (think this is as much personal preference as anything)

What platform was used for the WGS – I assume Illumina but don't see this specifically stated anywhere (apologies if I missed it). What database / programme was used to screen the WGSD data for chloramphenicol resistance genes?

How were the 42 isolates selected for analysis chosen? They clearly don't represent all the mismatched isolates – was the selection based on ST distribution? Or mix of carriage and clinical? ESBL-phenotype? Line 75 – I am not sure what you mean by control isolates – are these isolates without a phenotype/genotype mismatch. Suggest clarification.

Line 82: MIC results were within 2x of the breakpoint for CHL – is this meant to read within 2 doubling dilutions?

Line 83: We used the more sensitive MIC result to classify isolates as resistant or susceptible. I think this should read “We used the 2013 EUCAST MIC breakpoint of 8ug/mL to classify isolates as resistant or susceptible”

The fact that about 20% of the isolates had incorrect disc diffusion (compared to broth dilution) results is worth noting. While it doesn't detract from your overall findings, it may affect your initial data describing the proportion of isolates with genotype/phenotype mismatch (and maybe Fig 1 should include the fact that the phenotypic data in the fig is based on disc diffusion). More importantly, if the intention is to promote the use of chloramphenicol as a therapeutic option, reliable in-vitro AST results are important. Were the discrepancies VME, ME or minor errors? Have disc diffusion breakpoints changed since the initial testing that may explain this?

Table 1: Isolate CAE11U has an MIC of 12 – MICs are normally reported as doubling dilution values – should this MIC be 16? I don't think MG1655 should be in the table – the table is describing the clinical isolates and the discordance. MG1655 is part of the investigation into the discordance with cloned cat-variants. I think more appropriate in a separate table (or a table 1b).

Line 94/96 Table 1 – please clarify if the isolate in question is CAB119 or CAB199 (one of the 5 resistant isolates with no CAT activity). I found the section describing the resistant/ CAT negative isolates confusing to follow. You state that three isolates only contained catB4 – please list the three isolates. CAB199 (or 119?) carried floR and catB4. 13Y and 10H only had catB4 – not sure why this is repeated – could delete this from lines 97/98. 10A had catB4 and catA1. Why did you do the dCAT assay on only CAB119 and CAC13Y – or was it done on all 5 – not clear from results. Fig 2 shows more than 5 isolates that were phenotypically resistant (blue) but CAT negative - there are three that are resistant with no CAT activity and no gene. These are not mentioned?

Line 93 states that all S-isolates were negative for CAT activity regardless of cat-genes; lines 109-111 says much the same thing. Suggest you have a separate paragraph to address the phenotypically-Susc isolates. Maybe just a short summary of how many isolates were susceptible but had a cat gene, and which cat genes were commonest in these isolates. This would then lead in to the functional expression work more neatly.

Line 110 – maybe “suggesting” rather than “demonstrating”

Fig 2 – why do some isolates have more than one cat-absorbance result? Some have three, some 2 and some only one.

Line 132-138 – were the truncated catB3 sequences in Klebsiella and E coli isolates identical? There is no discussion / mention of differences or similarities across the two species.

Line 146 – From Supp Table 3, CAC10-A only has catA1. Suggest rephrase line 146 – I understood to mean all 5 isolates had the catB3-variant. In fact two have normal catB3, 2 have the truncated B3 and one only has catA1. Easier to then understand why you only aligned the contigs of the B3 and B3-variant isolates. Would it have been useful to include a couple of Klebsiella isolates for long read sequencing as well?

Line 233 – you investigated the link between clonal expansion and truncated catB3 in E coli. Could you not have done this for Klebsiella as well?

Line 292 – while the HRM assay will differentiate between functional and non-functional mutants in this study, it does not assess the presence of other resistance mechanisms (which you allude to earlier in the manuscript in lines 199-200. Phenotypic testing still has a place as long as it is reliable. And remember the disc diffusion showed about a 20% discordance with broth microdilution.

While I agree that there may be a place for chloramphenicol as directed therapy, I do have concerns about the side effect profile. This is discussed in the introduction but I think needs some more reflection in the discussion. I would imagine that one of the biggest barriers to the re-introduction of chloramphenicol is going to be the side effect issue. What can be done to mitigate this? In addition, if the agent does become more commonly used, resistance is likely to start emerging again and robust surveillance will be required. Given the limited lab capacity in many parts of Africa this will be a problem.

Comments from Reviewer #1:

The manuscript investigate the genetic basis of chloramphenicol (CHL) resistance in bacterial isolates, specifically *E. coli* and *Klebsiella pneumoniae*. It details the identification and functional analysis of chloramphenicol acetyltransferase (CAT) genes, which are responsible for CHL resistance in these bacteria. The study includes a comprehensive examination of various CAT gene variants, including *catA1*, *catA2*, *catB3*, and *catB4*, across different bacterial isolates.

The study reports on the presence of different CAT gene variants in *E. coli* and *Kp* isolates and their association with CHL resistance. Notably, it was found that the functional expression of *catB4* does not confer CHL resistance, suggesting that *catB4* is a non-functional variant. Further analysis revealed that *catB4* is a truncated version of *catB3*, with a significant portion of its sequence matching the IS26 transposase family. This truncation results in the loss of functional CAT activity, leading to the suggestion that *catB4* should be renamed to *catB3Δ443-633*. The study also explores the genomic context of the *catB3Δ443-633* variant, finding it commonly flanked by IS26 elements and part of an integron cassette. This suggests a mechanism for its spread and conservation across different bacterial isolates. The research identifies that the insertion of an IS5 element into the promoter region of *catA1* can reverse CHL resistance, highlighting the complex interplay between genetic elements in determining antibiotic resistance phenotypes.

A co-occurrence analysis among the isolates showed a positive relationship between *catB3Δ443-633* and other antibiotic resistance genes, indicating potential co-selection mechanisms. The analysis also revealed that *catB3* and *catB3Δ443-633* never co-occur in the same genome, suggesting a replacement or competitive exclusion relationship

Prevalence and Historical Emergence: The study investigates the prevalence and historical emergence of the *catB3Δ443-633* variant, finding it to be widespread and conserved, particularly in human isolates. This prevalence is contrasted with the wild type *catB3*, which is more common in non-human isolates, suggesting different selective pressures in different environments

The methods used to answer the research questions including disc diffusion and microbroth dilution, long-read sequencing, and functional assays to assess CAT activity and CHL resistance allowed for a detailed genetic and phenotypic characterization of the bacterial isolates.

Overall, the study provides significant insights into the genetic mechanisms underlying CHL resistance in *E. coli* and *Kp*, highlighting the role of CAT gene variants and other genetic elements in determining resistance phenotypes. The findings have implications for understanding antibiotic resistance mechanisms and could inform strategies for managing and mitigating resistance in bacterial pathogens.

We would like to thank the reviewer for taking the time to review our manuscript, the great summary of our work, the very positive feedback and helpful suggestions.

However, I have minor concerns at several aspects:

1. While the study highlighted the conservation and widespread occurrence of the truncated *catB3*Δ443-633 variant, it focused on a specific genetic context associated with other antimicrobial resistance (AMR) genes and IS26 elements. This limited scope may not fully capture the broader genetic landscape influencing CHL resistance mechanisms like overexpression of multidrug resistance efflux pumps (eg. AcrAB-TolC efflux).

We agree that our analysis focusses on the particular genetic context of the *catB3/4* and *catA1* variants, exploring mechanisms of inactivation of these genes. This approach was guided by the clinical and bacteriological context, where CHL resistance in disease causing isolates has been declining and hence mechanisms of re-emergence of CHL susceptibility are of interest.

We have therefore specifically investigated phenotypic-genotypic mismatches where the phenotype is sensitive, but genotype would predict resistance, and our analysis can explain 172/197 occurrences. We have deliberately not attempted to explore novel mechanisms of CHL resistance in a context of re-emerging susceptibility as this is beyond the scope of the research question and thus the paper.

We have amended the manuscript to acknowledge this and show a **new supplementary figure 1** as a heatmap showing presence of acquired CHL resistance genes and CHL resistance phenotype for our isolates. This shows that *catB4* and *catA1* are the dominant genes present in phenotypically susceptible isolates.

“Overall, the most prevalent *cat* genes were *catB4* (229) followed by *catA1* (189), *catA2* (104) and *catB3* (19) (Fig. 1c). Other known CHL resistance genes, *cmlA1* (29), *cmlA5* (23) and *floR* (14), were less common in the collection and their presence correlated well with phenotypic resistance (Supplementary Fig. 1). We selected a subset of 42 isolates, 27 *E. coli* (13 CHL-R, 14 CHL-S) and 15 *K. pneumoniae* (6 CHL-R, 9 CHL-S), based on a genotype-phenotype mismatch, i.e. CHL phenotypically susceptible isolates harbouring diverse *cat* genes and we included isolates without a genotype-phenotype mismatch with a matched *cat* gene as controls (same sequence type (ST) where available) for further in-depth functional analysis (Table 1) to investigate the molecular mechanism of those mismatches.” (Lines 70-79).

2. The co-occurrence analysis revealed positive relationships between *catB3*Δ443-633 and certain AMR genes but a negative relationship between *catB3* and *catB3*Δ443-633, indicating they do not co-occur in the same genome. However, this analysis may not account for all potential genetic interactions influencing CHL resistance.

We acknowledge that the analysis does not account for all potential genetic interactions on CHL resistance and have emphasised this more in the MS:

“Another limitation of our co-occurrence analysis is the limited set of (resistance) genes and isolates, which will not account for all genetic interactions influencing CHL resistance.” (Lines 322-324).

As discussed in our MS, our isolate collection is biased towards ESBL producers, and all larger genomic databases have their own biases (e.g. mostly pathogenic isolates from humans from the US and Europe) we thus think that a comprehensive analysis of genetic linkage is very challenging and beyond the scope of this manuscript. This also pertains to our answer to point 1 above – our focus was on re-emerging susceptibility.

3. The study emphasized the importance of integrating functional context into databases for accurate inference of resistance from genotypes. It suggested improvements in database curation to consider genetic contexts like promoters and transcription start sites, which are crucial for determining phenotypic resistance accurately. While this highlight is relevant, the same can also be applied to reference guide used to interpret phenotypic disc diffusion or MIC results. I believe this paper, is the best example and place to call for either a globally standardized breakpoint reference guidelines or specific breakpoints curated based on strains circulating at the nation, sub-regional, regional or African level.

There are already two globally acknowledged breakpoint setting organisations that continually review and update their guidance on phenotypic antimicrobial susceptibility testing, EUCAST and CLSI, however it would be futile to call for these organisations to merge and would therefore dilute the message of this manuscript. No such standards setting organisation exists as yet for genomic data.

Reviewer #2 (Remarks to the Author):

Thanks for the opportunity to review this very interesting paper. The authors compared phenotypic and genotypic markers of chloramphenicol resistance in E coli and Klebsiella, to understand discrepancies between the two susceptibility results. The findings are interesting both from a “basic science” perspective but also in better understanding some of the limitations of molecular resistance testing. The manuscript is well written although there are a few typos here and there and a careful read through is advised.

Many thanks for reviewing our MS. We are glad you found our findings interesting and relevant and thank you for your excellent suggestions which have strengthened our MS considerably. We have addressed each point below (and in the MS).

I have some comments and suggestions:

There is no statement regarding approval by an ethics review board, or of any form of waiver of ethics review.

We have now included ethical approval statements of both studies, where the bacterial isolates were originally obtained from. This has been added in a new section “bacterial isolates and genomes” on top of the methods section:

“The bacteraemia study was approved by the College of Medicine Research Ethics Committee (COMREC) of the University of Malawi (P.08/14/1614). The DASSIM study was approved by the Malawi College of Medicine (P.11/16/2063) and the research ethics committees of the Liverpool School of Tropical Medicine (16-062).” (Lines 360 – 363).

I think there should be more detail on the isolate collection – when were they collected (so how representative are they of current isolates). Were all the isolates ESBL-E? I thought suppl table 1 (Line 67) would contain details of isolates, but suppl table 1 appears to be a list of scripts. I do think some detail about the isolate collection would be useful.

We added a section in the materials and methods describing the isolates (see also response to comment below).

We apologise, the supplementary table 1 was in the folder of figure 1 – this has now been copied to the main directory of our GitHub repository.

I also found the methods slightly confusing as they make no reference to the isolates – all the isolate information is in the results. Given the structure of the manuscript (methods at the end) this is probably fine, but consider a small amount of information about the isolate collection in the methods (think this is as much personal preference as anything)

We added a section in the methods describing the isolates and genomes used in this study in more detail:

“Bacterial isolates and their genomes investigated in our study consist of 566 *E. coli* and 274 *K. pneumoniae* species complex (*KpSC*) isolates that had previously been isolated at Queen Elizabeth Central Hospital (QECH) in Blantyre, Malawi as part of two different studies. 164 (93 *E. coli*, 71 *KpSC*) isolates were from a sentinel surveillance study of bacteraemia and isolated between 1998 and 2016 [5]. 676 isolates (473 *E. coli*, 203 *KpSC*) were from a study investigating the ESBL-Enterobacterales colonisation in the gut of Malawian adults and collected in 2017 and 2018 (DASSIM study [14]).” (Lines 351 - 357).

What platform was used for the WGS – I assume Illumina but don't see this specifically stated anywhere (apologies if I missed it). What database / programme was used to screen the WGSD data for chloramphenicol resistance genes?

We updated the information:

Isolates were sequenced on the Illumina HiSeq X10 (Paired-end 150 bp) for the colonisation study¹⁴ and on the Illumina HiSeq 2000 (paired-end 100 b) for the bacteraemia study^{15, 16}. (Lines 357-359).

AMR genes from short-read data were called using ARIBA v.2.14.6⁵² and the curated ARG-ANNOT database used by SRST2²⁸. (Lines 470 – 471).

Abricate v0.0.9 and ResFinder was used for the co-occurrence analysis. (Lines 496-498).

How were the 42 isolates selected for analysis chosen? They clearly don't represent all the mismatched isolates – was the selection based on ST distribution? Or mix of carriage and clinical? ESBL-phenotype? Line 75 – I am not sure what you mean by control isolates – are these isolates without a phenotype/genotype mismatch. Suggest clarification.

Based on previously published genomic investigations of chloramphenicol resistance in *E. coli* and *KpSc*, we hypothesised that there would be only a few common AMR mechanisms, thus, we selected a subset of isolates including the most common mismatches (CHL-S catB4+ and CHL-S catA1+) for testing using functional assays. Once we had established the molecular mechanisms we were able to interrogate the genomes of all isolates in our collection. In the text below, we clarify how we selected the subset of 42 isolates and what we mean by control isolates.

“We selected a subset of 42 isolates, 27 *E. coli* (13 CHL-R, 14 CHL-S) and 15 *K. pneumoniae* (6 CHL-R, 9 CHL-S), based on a genotype-phenotype mismatch, i.e. CHL phenotypically susceptible isolates harbouring diverse *cat* genes and we included isolates without a genotype-phenotype mismatch with a matched *cat* gene as controls (same sequence type (ST) where available) for further in-depth functional analysis (Table 1) to investigate the molecular mechanism of those mismatches.” (Lines 74 – 79).

Line 82: MIC results were within 2x of the breakpoint for CHL – is this meant to read within 2 doubling dilutions?

We have changed this to ‘one doubling dilution’ to clarify.

Line 83: We used the more sensitive MIC result to classify isolates as resistant or susceptible. I think this should read “We used the 2013 EUCAST MIC breakpoint of 8ug/mL to classify isolates as resistant or susceptible”

Thanks for pointing this out, we have corrected the sentence as suggested.

The fact that about 20% of the isolates had incorrect disc diffusion (compared to broth dilution) results is worth noting. While it doesn't detract from your overall findings, it may affect your initial data describing the proportion of isolates with genotype/phenotype mismatch (and maybe Fig 1 should include the fact that the phenotypic data in the fig is based on disc diffusion). More importantly, if the intention is to promote the use of chloramphenicol as a therapeutic option, reliable in-vitro AST results are important. Were the discrepancies VME, ME or minor errors? Have disc diffusion breakpoints changed since the initial testing that may explain this?

Thanks for raising this important point. We acknowledge that we do not know the true nature of this discordance, especially as some of the isolates would have initially been tested by the Malawi Liverpool Wellcome Programme's routine bacteraemia service using British Society for Antimicrobial Chemotherapy breakpoints. We have addressed this in the MS by adding to the legend of Fig 1 that susceptibility it is based on disc diffusion and we state that the number of mismatches might have been over- or underestimated.

“For 81.0 % (34/42) isolates, microbroth dilution confirmed the result of disc testing and we concluded that phenotype-genotype mismatches were not explained by inaccurate phenotypic data, however, initial numbers of genotype-phenotype mismatches (Fig 1) might have been over- or underestimated. Five of the eight isolates that had discordant disc diffusion and MIC results were within 1 doubling dilution of the breakpoint for CHL (>8 µg/mL). We used the EUCAST (v.12.0) MIC breakpoint >8 µg/mL to classify isolates as resistant or susceptible. Four isolates were reclassified as susceptible and four as being resistant.” (Lines 81 – 88).

Table 1: Isolate CAE11U has an MIC of 12 – MICs are normally reported as doubling dilution values – should this MIC be 16? I don't think MG1655 should be in the table – the table is describing the clinical isolates and the discordance. MG1655 is part of the investigation into the discordance with cloned cat-variants. I think more appropriate in a separate table (or a table 1b).

MIC is 16. This has been corrected. We agree and have removed MG1655 expressing cat variants from Table 1. Results are summarised in results section. (Lines 120 – 126).

Lie 94/96 Table 1 – please clarify if the isolate in question is CAB119 or CAB199 (one of the 5 resistant isolates with no CAT activity). I found the section describing the resistant/ CAT negative isolates confusing to follow. You state that three isolates only contained catB4 – please list the three isolates. CAB199 (or 119?) carried floR and catB4. 13Y and 10H only had catB4 – not sure why this is repeated – could delete this from lines 97/98. 10A had catB4 and catA1.

Why did you do the dCAT assay on only CAB119 and CAC13Y – or was it done on all 5 – not clear from results.

Fig 2 shows more than 5 isolates that were phenotypically resistant (blue) but CAT negative - there are three that are resistant with no CAT activity and no gene. These are not mentioned?

The isolate is CAB199 and was wrongly labelled as 119 – this has been corrected.

We have labelled the three isolates where catB4 was the only cat gene and removed the repetitive sentence about CAC13Y and CAC10H:

“Three of those CHL-R/rCAT negative isolates (CAB119, CAC10H, CAC13Y) carried *catB4* genes and this was the only *cat* gene present. Isolate CAB199 (CHL MIC = 128 µg/mL) co-carried a *floR* gene which could contribute to the CHL-R phenotype. CAC10A had a *catA1* in addition to *catB4* but a weak CHL-R phenotype (MIC = 16 µg/mL).” (Lines 100 -104).

We tested the D-cat assay with the two isolates stated and since it became clear that *catB4* was non-functional we have not repeated the assay with all 5 isolates.

We don't know the mechanism of resistance in the 3 resistant isolates without an apparent acquired CHL gene and did not further investigate this as our primary focus was to understand the mismatches of phenotypically susceptible isolates carrying CHL resistance genes.

Line 93 states that all S-isolates were negative for CAT activity regardless of cat-genes; lines 109-111 says much the same thing. Suggest you have a separate paragraph to address the

phenotypically-Susc isolates. Maybe just a short summary of how many isolates were susceptible but had a *cat* gene, and which *cat* genes were commonest in these isolates. This would then lead in to the functional expression work more neatly.

Thank you for this suggestion: we separated a short paragraph summarising the CHL-S isolates.

“Of 23 phenotypically susceptible isolates, 15 carried *cat* genes (3 with *catA1*, 2 with *catA1* + *catB4*, 10 with *catB4*), none of these showed CAT activity, suggesting that *cat* genes in those CHL-S isolates are not expressed.” (Lines 114 – 116).

Line 110 – maybe “suggesting” rather than “demonstrating”

Fig 2 – why do some isolates have more than one *cat*-absorbance result? Some have three, some 2 and some only one.

We agree and replaced ‘demonstrating’ with ‘suggesting’.

For each isolate, the rCAT assay has been performed in duplicate. Some isolates have additional repeats, to confirm the positive signal. Raw data is accessible on our linked GitHub in the respective folder of Fig 2.

Line 132-138 – were the truncated *catB3* sequences in *Klebsiella* and *E. coli* isolates identical? There is no discussion / mention of differences or similarities across the two species.

The truncation of *catB3* is identical irrespective of species. As shown in Suppl. Fig3 b (previously supplementary Fig2 b) there are some smaller truncations of *catB3*, however, those would not have been called as *catB4*. We added a sentence to the MS: “... and there is no difference between *E. coli* and *KpSC*” (Lines 141-142).

Line 146 – From Supp Table 3, CAC10-A only has *catA1*. Suggest rephrase line 146 – I understood to mean all 5 isolates had the *catB3*-variant. In fact two have normal *catB3*, 2 have the truncated B3 and one only has *catA1*. Easier to then understand why you only aligned the contigs of the B3 and B3-variant isolates. Would it have been useful to include a couple of *Klebsiella* isolates for long read sequencing as well?

We clarified this in the MS: “We selected five *E. coli* isolates for long-read sequencing to investigate the genomic context of *catB3*, *catB3* $\Delta^{443-633}$ and *catA1* (Supplementary Table 3).” (Lines 152 – 153).

We sequenced several *Klebsiella* isolates on long-read but unfortunately there were some quality issues and genomes were fragmented and could not be resolved.

Line 233 – you investigated the link between clonal expansion and truncated *catB3* in *E. coli*. Could you not have done this for *Klebsiella* as well?

We leveraged an existing and well-curated *E. coli* collection to do this analysis, and, to the best of our knowledge, we did not find an *Klebsiella* collection equal in size and quality. We

acknowledge that this is a limitation to comprehensively understand clonal expansion of the truncated *catB3* in *Klebsiella*.

Adjusted in MS: “We expanded our analysis and leveraged an existing collection of 10k genomes consisting of the top 100 common STs in *E. coli* (previously curated and downloaded from Enterobase (2020)[36] where each ST was randomly sampled to select 100 genomes.” (Lines 243 – 245).

Line 292 – while the HRM assay will differentiate between functional and non-functional mutants in this study, it does not assess the presence of other resistance mechanisms (which you allude to earlier in the manuscript in lines 199-200. Phenotypic testing still has a place as long as it is reliable. And remember the disc diffusion showed about a 20% discordance with broth microdilution.

The HRM was previously developed to capture known acquired CHL resistance genes in Enterobacterales (*cat* genes, *floR*, *cmlA*) and we expanded the primer set to distinguish between the functional and non-functional *cat* variants described in our study. We agree that phenotypic AST is still the gold standard, and we believe our study highlights the current limitations with genomic or molecular testing for phenotypic resistance clearly. We also believe though that molecular testing has potential to be faster and cheaper than culture-based AST. We amended our statement in the MS to clarify and highlight limitations. “In the Malawian isolates the gold-standard AST correctly determined phenotypic CHL susceptibility, however, rapid molecular diagnostics have the potential to be faster and low cost, but require knowledge of resistance mechanisms. We applied the HRM assay, previously developed to detect acquired CHL resistance genes [33], to distinguish between functional and non-functional *cat* genes found in our study.” (Lines 304 – 308).

While I agree that there may be a place for chloramphenicol as directed therapy, I do have concerns about the side effect profile. This is discussed in the introduction but I think needs some more reflection in the discussion. I would imagine that one of the biggest barriers to the re-introduction of chloramphenicol is going to be the side effect issue. What can be done to mitigate this? In addition, if the agent does become more commonly used, resistance is likely to start emerging again and robust surveillance will be required. Given the limited lab capacity in many parts of Africa this will be a problem.

We agree that CHL reintroduction will need to be balanced carefully given its potential toxicity. Since CHL has been used as a first line agent in Malawi prior to 2004 we believe that there is considerable local experience administering CHL. We have expanded on these issues in the discussion:

“To effectively re-introduce CHL as a reserve treatment option for Enterobacterales infections confirmed as 3GC-resistant, we must consider the potential for reversion to CHL-resistance. Isolates carrying IS5-*catA1* could potentially rapidly revert to high-level resistance since the *catA1* CDS is still present in the genome. Our data using experimental evolution suggest that this is not the case and selection with CHL did not lead to expression of *catA1*. One of the evolved isolates co-carried a truncated *catB3* and we expected no reversion to a functional *catB3* upon CHL selection because the missing 3' prime end of the gene is no longer present

in the genome. Indeed, no functional cat emerged. Resistance to CHL will ultimately re-emerge when CHL is used more frequently but recent efforts of antibiotic stewardship in Malawi aim to keep selection pressures low. Critically, CHL was once the empirical intravenous agent for the management of sepsis across much of Africa and was available in the community, we are proposing limited re-introduction for blood culture confirmed infection, which result in orders of magnitude less use.” (Lines 277 – 289).

...

“several studies have reported high or increasing rates of phenotypic CHL susceptibility among MDR Gram-negative bacteria suggesting CHL as a viable alternative treatment option in those settings [42-44]. Further, CHL is an affordable and useful antimicrobial in terms of bioavailability, tissue penetration and broad spectrum of action [45, 46]. However, the prevalence of CHL susceptibility and the rare but severe side-effects of CHL, preclude the use of CHL for empirical management of sepsis in our setting: we instead envisage CHL to be used in critically ill adult (CHL is contraindicates for children given its toxicity) and hospitalised patients with confirmed ESBL-E and CHL-S infection as a reserve agent when no treatment alternative is available. This has the added advantage of keeping selection pressure for CHL resistance low but does require rapid determination of CHL susceptibility phenotype.” (Lines 294 – 303).

...

“Lastly, our data support the reintroduction of CHL, carefully balancing its potentially severe side effects, as a last-line treatment option for patients critically ill with ESBL-E infections in Malawi and similar settings.” (Lines 346 - 348).